# Postpartum Depression: Etiology, Treatment, and Consequences for Maternal Care

**DOI:** 10.3390/diagnostics14090865

**Published:** 2024-04-23

**Authors:** Daiana Anne-Marie Dimcea, Răzvan-Cosmin Petca, Mihai Cristian Dumitrașcu, Florica Șandru, Claudia Mehedințu, Aida Petca

**Affiliations:** 1Department of Obstetrics and Gynecology, “Carol Davila” University of Medicine and Pharmacy, 050474 Bucharest, Romania; daiana-anne-marie.dimcea@drd.umfcd.ro (D.A.-M.D.); mihai.dumitrascu@umfcd.ro (M.C.D.); claudia.mehedintu@umfcd.ro (C.M.); aida.petca@umfcd.ro (A.P.); 2Department of Urology, “Carol Davila” University of Medicine and Pharmacy, 050474 Bucharest, Romania; 3Department of Urology, “Prof. Dr. Th. Burghele” Clinical Hospital, 050659 Bucharest, Romania; 4Department of Obstetrics and Gynecology, University Emergency Hospital, 050098 Bucharest, Romania; 5Department of Dermatology, “Carol Davila” University of Medicine and Pharmacy, 050474 Bucharest, Romania; florica.sandru@umfcd.ro; 6Department of Dermatology, Elias University Emergency Hospital, 011461 Bucharest, Romania; 7Department of Obstetrics and Gynecology, Filantropia Clinical Hospital, 011171 Bucharest, Romania; 8Department of Obstetrics and Gynecology, Elias University Emergency Hospital, 011461 Bucharest, Romania

**Keywords:** postpartum depression, screening, postpartum depression treatment, obstetrician’s role

## Abstract

Postpartum depression (PPD) is a disabling condition that has recently shown an increase in prevalence, becoming an essential public health problem. This study is a qualitative review summarizing the most frequent risk factors associated with PPD, evaluating molecular aspects of PPD and current approaches to detect and prevent PPD. The most prevalent risk factors were detected in the areas of economic and social factors, obstetrical history, lifestyle, and history of mental illness. Research on the genetic basis for PPD has taken place in recent years to identify the genes responsible for establishing targeted therapeutic methods and understanding its pathogenesis. The most frequently studied candidate gene was the serotonin transporter gene (SERT) associated with PPD. Among biological studies, antidepressants and psychological interventions provided the most evidence of successful intervention. The obstetrician can serve an essential role in screening for and treating PPD. Postpartum women with risk factors should be screened using the Edinburgh Postnatal Depression Scale (EPDS), but, at the moment, there are no prevention programs in Europe. In conclusion, data from this review increase concerns among this vulnerable population and can be used to design a screening tool for high-risk pregnant women and create a prevention program.

## 1. General Features

Depression is one of the most disabling conditions for women of childbearing age. For women worldwide aged 15–44, after HIV/AIDS, it is the second leading cause of total disability [1,2].

Postpartum depression (PPD) (also known as peripartum depression or major depressive disorder with peripartum onset) is defined according to DSM-5 diagnostic criteria as a depressive episode that begins during pregnancy or the first four weeks after birth; however, women remain at risk of developing depression several months after childbirth [3]. In the case of some women, a recurrence of depression occurs after birth, while other patients experience the first symptoms in the postpartum period [4].

Prior studies have shown that the worldwide prevalence of PPD until 2017 would range from 9.5% in high-income countries to 20.8% in middle-income regions and around 25.8% in low-income nations [5,6].

A systematic review of 291 studies from 56 countries conducted in order to report PPD prevalence found a global pooled prevalence of PPD of 17.7% (95% CI: 16.6–18.8%), highlighting the significance of addressing this condition as a critical public health concern [7]. These estimates are similar to the 19% prevalence for PPD derived from studies of relatively low- and middle-income countries [8]. In India, postnatal depression is a frequent occurrence, and, according to recent statistics, the incidence is between 19.8% and 23.3% [9]. In developing countries, such as Romania, the prevalence of PPD cases is the highest [20.14 (range: 16.39–24.50)] compared to upper-middle and high-income countries [10]. According to the largest meta-analysis of PPD to date, the global prevalence of postnatal depression is 17.22% (95% CI: 16.00–18.51) [11]. Moreover, the results of the studies that analyzed the prevalence of postnatal depression during the COVID-19 pandemic indicate a two-fold higher incidence of postnatal depression cases compared to the non-pandemic period [12].

Untreated PPD has a negative impact on both mother and infant. Studies have indicated the risk to children of untreated depressed mothers (compared to mothers without PPD), including problems such as poor cognitive functioning, behavioral inhibition, emotional maladjustment, violent behavior, externalizing disorders, and psychiatric disorders with onset in adolescence [13].

Despite the increase in PPD incidence in recent years, there are multiple barriers to the provision of optimal clinical care for women with this condition. The majority of women decline to seek professional assistance due to the social stigma associated with mental illness. Finally, doctors may fail to catch the onset of depressive symptoms or consider them insignificant, which leads to underdiagnosis or undertreatment of the mental condition [4].

The aim of this review was to cover a broad range of issues in PPD, such as risk factors, with particular attention to ones that may be useful to identify at-risk pregnant and postpartum women; psychiatric diagnosis and biological diagnosis; clinical manifestations of PPD; genetic background associated with PPD; psychotherapy and pharmacotherapy trials, with an emphasis on therapy modifications that are specific to PPD; the use of antidepressant medication for breastfeeding mothers; the role of obstetrical–gynecologist specialists in the detection women at risk for PPD; and the public health context, with a particular emphasis on screening and management in primary care.

## 2. Epidemiological Factors

There is evidence that biological factors, such as hormonal factors, genetics, and immune function, among other types of causes, play an essential role in triggering PPD [14].
(a).The following demographic risk factors can be grouped according to the strength of association with PPD: depression and anxiety in pregnancy, history of depression, excessive stress caused by life events, poor marital relations, lack of social support, and low self-esteem are strongly associated with PPD [15]. On the opposite side, low socioeconomic status, single marital status, and unwanted pregnancy are considered to have a weaker association with PPD [16].(b).Regarding obstetrics risk factors, Mayberry et al. reported a higher predisposition of depressive symptomatology in multiparous patients compared to nulliparous [17]. In a study conducted by Mathisen in 2013 that studied 86 mothers in the first six weeks postpartum, it was shown that women with two or more children are associated with a higher risk of the onset of depressive symptoms because of psychological distress [18]. Furthermore, the following are also considered obstetrical risk factors: a high-risk pregnancy requiring cesarean section, perinatal complications, and incapacity to breastfeed [18].(c).PPD may also be associated with different sensitivity to hormonal fluctuations [19]. A study conducted by Trifu S. et al. showed that the dramatic drop in the progesterone hormone after birth may have a role in PPD. A possible explanation is represented by the association between its reduction and decreased irritability. Furthermore, estradiol hormone levels decrease in order to stimulate lactation. The decreased estradiol level deprives the body of a natural defense to fight against depression while having an essential role in serotoninergic transmission by enhancing serotonin synthesis [20]. In addition to sensitivity to estrogen and progesterone fluctuations, biological theories have demonstrated that other changes, such as those of gonadal hormones but also neuroactive steroid levels after birth, altered cytokines and hypothalamic–pituitary–adrenal (HPA) axis hormones fluctuations, acid-altered fats, and oxytocin and arginine vasopressin levels are involved in the production of PPD onset in predisposed women [21,22]. The involvement of the serotoninergic system was suggested by other studies that evaluated altered platelet serotonin transporter binding and decreased postsynaptic serotonin 1-A receptor binding in the anterior cingulate cortex and mesial temporal cortex [23,24]. Protein-enriched foods reduce tryptophan and serotonin levels in the brain, while carbohydrates have an antagonistic effect. In nutritional deficiencies, low doses of tryptophan (a serotonin precursor) increase the rate of development of postpartum depressive symptoms [25]. Oxytocin also plays an important role in emotion regulation and higher doses of oxytocin in the second trimester of pregnancy were predictors of postnatal depression in the first two weeks after birth [26].(d).Social support refers to emotional support, intelligence support, and empathic relationships. Reducing social support is the most important environmental factor in the onset of PPD and anxiety. Husband abuse and other forms of domestic violence during pregnancy are seen as contributing factors to the increase in the incidence of PPD [27]. Another social factor is employee status. Mostly, women with professional careers are thought to be associated with a reduced risk of PPD [28]. Based on the finding from Lewis et al.’s study that evaluated the relationship between employment status and depression symptomatology among women at risk for PPD, it was demonstrated that postpartum women who are employed were less likely to report higher depression symptomatology than unemployed women. The study found that the protective effect of employment on PPD also occurs in women who are at an increased risk of depression [29].(e).Multiple lifestyle factors have been associated with the risk of depression in general, including substance abuse, smoking, nutrition, sleep, physical activity, or vitamin D deficiency [30]. Findings from the GUSTO cohort published in 2020 that evaluated the cumulative risk of lifestyle behaviors on depressive symptoms during pregnancy and after delivery have shown that women with at least four risk factors had at least a sixfold higher prevalence of having depression compared to those with zero or one risk factor. Sleep appeared to be the most substantial contributor to depressive symptoms, according to statistical analysis, as women may experience difficulty sleeping due to normal changes of pregnancy in their bodies. On the opposite level, vitamin D concentrations and MET minutes of physical activity contributed the least to the variance of an analytical sample [31].

Biological factors and social factors create intertwined rings that make women susceptible to PPD by affecting each other. Furthermore, many environmental factors, such as socioeconomic factors, cause crisis conditions and PPD by influencing mental health during pregnancy [32].

## 3. Gene Expression Profiles: Molecular Aspects of PPD

Research on the genetic basis for PPD has taken place in recent years to identify the genes responsible for establishing targeted therapeutic methods and understanding its pathogenesis [33]. Recent studies have focused on this issue and support the existence of an underlying genetic vulnerability to PPD, but it is still quite inconclusive whether this vulnerability is different from or correlated with the genetic susceptibility of other psychiatric conditions [34,35]. A recent study conducted in Sweden on 3.427 twin patients and over 500.000 sisters concluded that the heritability in PPD is 54% for the first category and 44% for the second category [36].

Most of the molecular genetic studies on PPD have focused on candidate genes. These studies have selected and tested a small number of candidate genes or genetic variants for association with a phenotype. These candidates were chosen based on the assumption that the gene products influence the phenotype through a hypothesized biological mechanism [37,38]. A number of candidate gene studies have been conducted in PPD, but it is considered that most of the candidate gene studies are thought to play a role in major depressive disorder (MDD) rather than PPD [33].

The most frequently studied candidate gene was the serotonin transporter gene (*SERT*), which presents two primary polymorphisms, of which the *5-HTTLPR* polymorphism is the most frequently associated with PPD. There is evidence for the role played by the *SERT* gene in PPD; however, studies show variable outcomes [25,39]. Studies evaluating *5-HTTLPR* have demonstrated a positive pattern when PPD is measured in the immediate postpartum period (up to 8 weeks) and a negative pattern when measured at a larger interval [33]. A recent study conducted on 276 postpartum women in Brazil that evaluated the role of serotonin transporter gene polymorphism (*5-HTTLPR*) and stressful life events regarding the risk of PPD symptoms found similar results as a study conducted by Sanjuan et al. that showed that depressive symptoms were associated with high-expression *5-HTT* genotypes only in the early postpartum period. The authors suggested that the rapid uptake of serotonin in L allele carriers, combined with the reduced availability of brain tryptophan during the postpartum period, could increase depressive symptoms during this time [25,39].

The catechol-O-methyltransferase (*COMT*) gene codes for enzymes that break down catecholamine neurotransmitters, including epinephrine, norepinephrine, and dopamine. It has been studied before for its association with stress, depression, and anxiety. Still, unfortunately, results obtained in different studies were inconsistent [33]. In a study conducted on 116 women of Brazilian–Caucasian descent, it was found that *Val158Met COMT* polymorphism was associated with postpartum depressive symptoms. According to the same authors, this polymorphism has been found to be associated with both major and bipolar depression; but, also, many negative results were published, rendering any conclusion hazardous [40].

Oxytocin plays an essential role in regulating emotions, social interactions, and stress. A recent study has shown a significant interaction between the *rs53576* genotype and the presence of prenatal depression on PPD (*p* = 0.0081), and further details showed that women who do not display depression in pregnancy but who harbor the *rs53576_CG* genotype are three times more likely to develop PPD in comparison to women with lower methylation levels. This was the first investigation of the oxytocin receptor (*OXTR)* as a potential clinical genetic biomarker associated with PPD and further research is required. The biologically at-risk women in this study did not display elevated symptoms of depression in pregnancy but went on to display an increased risk of PPD after birth [41].

Estrogen receptor (*ESR*) genes are considered candidate genes for PPD, and, from its subtypes of estrogen receptors, only *ESR 1* has been studied in relation to PPD. In a pilot study published in 2013 by Pinsonneault J.K. et al., association analysis of PPD and polymorphisms in *ESR1* showed that two variants in the *ESR1* gene, the *TA repeat* (*p* = 0.07) and *rs2077647* (*p* = 0.03), were significantly associated with EPDS scores in 156 postpartum women. One of the limitations of the study was a notably small sample size, which led to low power and needs to be replicated in larger samples [42]. Furthermore, *ESR2* has not been studied in PPD, and knowing its role in estrogen signaling may also be productive [33].

The monoamine oxidase A (*MAOA)* gene codes for enzymes that break down amine neurotransmitters, including serotonin, dopamine, and norepinephrine. During the last 15 years of work, promising studies have shown a positive association between *MAOA* gene polymorphism and PPD. Two of these studies were positive at six weeks postpartum and negative at 3 and 6 months postpartum [43,44]. In the first study, the main finding was the association between PPD symptoms six weeks after delivery and the *COMT-Val^158^* Met polymorphism alone and in interaction with functional monoaminergic polymorphisms and environmental stressors. The *COMT-Met^158^* allele has been associated with major and bipolar depression, adverse response to antidepressant treatment, anxiety disorders, and brain activation elicited by aversive stimuli [43].

The *AKR1C2* (aldo-keto reductase family 1) gene encodes *AKR1C2* (aldo-keto reductase family 1, members C2) and influences the relative levels between progesterone and allopregnanolone in humans by regulating the synthetic pathway from progesterone to allopregnanolone, which may be intrinsic to the hormonal fluctuations in patients with PPD. Women with the AA genotype of *SNP rs1937863* at *AKR1C2* gene are associated with lower allopregnanolone levels and lower depression scores, suggesting that *AKR1C2* variants may alter susceptibility to depressive symptoms by affecting central progesterone synthesis, which is not a major factor in the development of PPD [45]. However, the *AKR1C2* gene can be used as a candidate gene to study the level of progesterone and allopregnanolone, AKR1C2 gene polymorphism, and PPD.

In recent years, there have been multiple genes extensively studied to demonstrate an association with PPD, such as the brain-derived neurotrophic factor (*BDNF*) or hemicentin-1 gene (*HMNC1*), which have shown a modest association [46,47]. On the basis of these data, the *BDNF Met allele* is associated with an increase in anxiety-like behavior in females that emerges during the period of sexual maturity. It has been suggested that women carrying the *BDNF Met allele* may be more sensitive to the potential impact of reproductive hormones on anxiety and depressive disorders [46]. The gene *HMCN1* may contain polymorphisms that confer susceptibility to postpartum mood symptoms, being particularly expressed in the hippocampus, a brain region involved in depression. Twenty-seven single nucleotide polymorphisms (*SNPs*) in *HMCN1* were significant (*p* < 0.05), suggesting an association with PPD, but future studies replicating these findings are mandatory [47].

The genetic risk for PPD may have a component that overlaps with the genetic risk for major depressive disorder (MDD) and/or bipolar disorder and, in addition, a component that is specific to PPD itself. Candidate gene association studies and heritability studies indicate a more substantial genetic basis when PPD occurs in the early days after birth. Further exploration of PPD-associated genes is justified and requires more extensive studies in the coming years. Ultimately, the identification of the genetic underpinnings of this disorder may bring into light the biological basis for mood disorders more generally (Table 1) [33].

## 4. Clinical Presentation of PPD Onset: Diagnostic Criteria and Screening for Women at Risk of PPD

According to the Diagnostic and Statistical Manual of Mental Disorders, fifth edition (DSM-5; 2013), PPD occurs when a person meets the criteria for a major depressive episode during the first four weeks postpartum; however, women are at risk of developing depression several months after birth [3,4]. Some patients show a recurrence of depression after delivery, while others develop the first symptoms in the postnatal period [3]. Risk factors for PPD include depression during pregnancy, anxiety during pregnancy, stressful life events during pregnancy or the early puerperium, low levels of social support, and a personal or family history of depression [4].

The diagnosis of major depressive syndrome requires the patient to meet at least five of the characteristic symptoms occurring on several days over a period of at least two weeks and producing significant interference or distress in daily life [4,48]. A general status altered by the use of substances (such as drugs, alcohol, or medication) or that is the result of a medical condition is not considered a criterion for the diagnosis of a major depressive episode.

The signs and symptoms of PPD are identical to those of non-puerperal depression, with the exception that the former is associated with a history of childbirth [49].

PPD is differentiated from two other entities of emotional impairment in the postnatal period mentioned in DSM-IV. The postpartum blues (often called “baby blues”) represents a transient and mild behavioral change that begins in the first week postpartum and lasts from a few hours to a few days. Characteristic symptoms include uncontrollable crying, psycho-emotional lability, anxiety, and insomnia. Between 50% and 80% of postnatal patients experience the blues, but the consequences are minimal. In comparison, patients suffering from postnatal psychosis experience symptoms such as confused thinking, disillusionment, hallucinations, and disorganized thinking. The prevalence of postpartum psychosis is between 1 in 500 cases and 1 in 1.000 patients. The duration of postpartum psychosis varies and requires rapid diagnosis and hospitalization. According to DSM-5, all mood and psychotic symptoms occurring during pregnancy or within the first four weeks following delivery are referred to as peripartum mood or psychotic episodes [3,48].

A meta-analysis conducted by Gavin et al., who diagnosed patients with PPD employing the structured interview, reported that the combined prevalence of major depressive episodes varies between 6.5% and 12.9% in the first 6 months postpartum, with a peak incidence between 2 and 6 months after delivery [50]. Another cohort study conducted in Denmark reported that the first three months after birth present an increased risk of the onset of psychiatric conditions in primiparous patients [51].

These findings emphasize the need for obstetricians to assess patients for psychiatric history correctly and, with the help of psychiatrists, optimize the treatment of mothers in the peripartum period [4].

Women who are at risk of postnatal depression should be identified as early as possible in pregnancy so that assessment and treatment can be initiated promptly [52].

Over recent years, there has been an increase in focus on the importance of the early detection and treatment of depression during pregnancy. Screening can be performed four to six weeks in the postpartum period. Several methods of detecting depression symptoms among women have been tested [52].

During the history taking of the patient by the obstetrician, special attention should be paid to the personal or family history of depression, postpartum psychosis, or bipolar disorder. Once depressive symptoms have been identified, a comprehensive evaluation of risks that influence the clinician’s treatment recommendations should be performed. Furthermore, patients should be assessed for social support, substance use, or partner abuse. The evaluation of patients should include a complete mental status examination and a physical examination to determine if the symptoms suggest a medical cause. In this context, laboratory investigations (including dosages of hemoglobin and thyroid hormones) are necessary [53].

Patients with risk factors should be screened with the Edinburgh Postnatal Depression Scale (EPDS) [52]. This validated instrument consists of 10 items and has a completion time of approximately 5 min. It also includes a question about suicidal ideation proposed by Cox et al. in 1987 [52]. Each question is scored on a scale of zero to three. Established cut-off values (a score greater than 10 is suggestive of a mild depressive episode and a score of 13 or more is suggestive of a moderate or severe depressive episode) allow clinicians to identify women at increased risk of PPD requiring additional clinical evaluation [54]. The best time to screen for PPD with this tool is in the first month after giving birth. A study conducted in Hungary between 2010 and 2011 that applied this tool to assess patients who gave birth in the last 6 to 8 weeks showed that the application of this scale demonstrated good reliability and internal consistency after applying statistical analysis [55].

The Postpartum Depression Screening Scale (PDSS) is also used as a screening tool for the detection of PPD, evolving from qualitative interviews to exploring the postpartum maternal experience. The first part comprises seven items; patients with a PDSS score ≥ 14 receive an extended questionnaire with another 28 items. A score ≥ 60 suggests a risk of a minor or major depressive episode; a score ≥ 80 is predictive of a major depressive episode. The PDSS scale has been used effectively in telephone assessment but presents an increased risk of false positive results that restrict its usefulness [52,56].

The Patient Health Questionnaire (PHQ-9) is a 9-item scale with two components used in the analysis of symptoms and functional imbalance in the diagnosis of depressive syndrome. It is also used to make a severity score to assess the effectiveness of treatment. The total score can range from 1 to 27 and can be classified from mild to severe forms of depression. The high sensitivity (88%) and specificity (88%) of the method ensure validity for the identification of risk factors [56].

The Edinburgh Postnatal Depression Scale (EPDS) and Postpartum Depression Screening Scale (PDSS) are used specifically for the diagnosis of postnatal depression, while the Patient Health Questionnaire (PHQ-9) is recommended for the diagnosis of depression in psychiatric medical institutions [54].

In a study conducted by Chadka-Hooks et al. in which the original Edinburgh Postnatal Depression Scale (EPDS) was used to assess screening practices in the healthcare system (including obstetricians, pediatricians, and family physicians) to assess familiarity with PPD screening methods, it was shown that healthcare providers are not familiar with screening methods [57].

Acceptability for the assessment of PPD is an important indicator of the likelihood that patients will respond to questionnaires to inform about their risk of depression [56]. In a study led by Matthey S. et al., the researchers showed that patients who readily agreed to complete the Edinburgh Postnatal Depression Scale (EPDS) had a lower risk of PPD. Thus, in the case of 87% of the studied group, no discomfort was felt in completing the questionnaire; these patients had EPDS scores < 13, compared to 64% of women at high risk and with EPDS scores ≥ 13 (chi-square = 31.9, df = 2, *p* < 0.0001) [58]. Therefore, the perception of discomfort with PPD screening is closely related to the risk of developing depression [56].

Although screening for the detection of perinatal depression is becoming more and more accepted both in the United States of America and in Europe, there are controversies regarding the possibility of overidentifying patients at risk or over-pathologizing behavioral symptoms in the postpartum period or the inability of the system to provide the necessary treatment and follow-up. These issues are currently unresolved. However, in recent years, much work has been conducted to develop new therapies or adapt existing ones used in PPD. Thus, current studies have demonstrated the importance of screening methods for the early detection of PPD [59,60].

## 5. Alternative Predictors Used in PPD Diagnosis

The diagnosis of PPD can be performed in multiple ways. The DSM-5 is used to guide an interview design, and self-report tools, such as questionnaires, have been extensively employed in the clinical evaluation of PPD. However, using only such arbitrary scales could result in subjective bias. In order to improve diagnostic accuracy, the most efficient method in clinical practice is to apply some objective markers in addition to these subjective assessments [61]. Consequently, the endocrine system profile is considered, together with a self-rating scale, clinic interviews, and hormones of the hypothalamic–pituitary–adrenal (HPA) axis [62]. In PPD, there are three important HPA axis hormones that have been investigated because of the psychological changes mothers experience: the corticotrophin-releasing hormone (CRH), the adrenocorticotropic hormone (ACTH), and cortisol [63,64].

During pregnancy, childbirth, and lactation, the HPA axis and level of hormones modify severely, especially in the last weeks of pregnancy when cortisol levels rise most steeply, reaching levels three times higher than in non-pregnant women [65]. Rather than the HPA axis, the placenta is principally responsible for the higher basal cortisol concentrations during pregnancy. The placenta progressively takes over the function of the endocrine gland throughout pregnancy. In addition to producing progesterone and estrogen, the placenta also produces pCRH, which shares structural and bioactive similarities with hypothalamic CRH [66]. Cortisol levels rise during parturition, more noticeably after vaginal delivery. The difficulty of labor may contribute to an increase in cortisol levels. After delivery, plasma CRH concentrations return to preconception levels within 15 h. [67]. Women with significant depression had reduced cortisol stress responses. A study conducted by O’Connor et al. demonstrated that a diagnosis of depression in mothers was significantly associated with cortisol levels. The results from this study indicate that the strongest effect was that women with a diagnosis of depression had a lower initial waking level (depression × wakeup) [68].

Progesterone level and PPD have a strong connection. It has been demonstrated that the level of allopregnanolone increased steadily during gestation and then sharply decreased following birth [69]. A study conducted by Osborne L.M. et al. showed an association between a low level of allopregnanolone during the second trimester and higher depressive symptoms at six weeks postpartum, although it did not reach statistical significance in the adjusted model [70]. Also, lower levels of oxytocin are linked to an increased risk of PPD development in both gestational and postpartum periods. Jobst A. et al. demonstrated in one study that oxytocin plasma levels decreased from late pregnancy to the time of delivery in the depressed group, whereas oxytocin plasma levels increased continuously from pregnancy to the postpartum period in the non-depressed group. This difference in the course of oxytocin levels was significant and predicted postpartum depressive symptoms [71].

There is evidence that the thyroid hormone, which has been associated with severe neurological impairments, may enhance the risk of PPD due to its aberrant expression in the early postpartum period [72]. There is a positive association between a high thyroid stimulating hormone (TSH) (defined as >4 mIU/L) with free thyroxine (fT4) within the normal pre-pregnancy range at delivery and the development of self-reported depressive symptoms six months postpartum [73]. The presence of thyroid peroxidase antibodies (TPO-ab) during early gestation is predictive of both autoimmune thyroid disorders and depression. A positive TPO-ab status was associated with an increased risk for first-onset depression at four months postpartum (adjusted OR: 3.8; 95% CI: 1.3–11.6), but not at other postpartum time points [74].

It has been demonstrated that in the third trimester of pregnancy, proinflammatory cytokine levels are significantly greater, and these women are also at risk of developing PPD. The presence of a proinflammatory state throughout late pregnancy and the early postpartum period is essential to increase the risk of PPD [75]. Elevated levels of the potent proinflammatory cytokine interleukin 1β (IL-1 β) very early in the postpartum period increase a woman’s risk of developing symptoms of depression [76]. Furthermore, the serum levels of IL-6 after delivery were significantly higher in women with PPD. The optimal cut-off value as an indicator for screening was estimated to be 24 pg/mL, which yielded a sensitivity of 83.1% and a specificity of 79.4% [77].

Identifying biochemical and dietary indicators for PPD diagnostics has recently gained more attention. A significant finding of one of the recent studies is that lowered zinc levels during pregnancy are associated with prenatal and PPD symptoms [78]. PDD was also related to lower 25-hydroxylated vitamin D (25OHD) levels (≤80 nmol/L) in one study [79].

The transition to motherhood, whether for the first time or subsequently, involves a series of anatomical and physiological changes during pregnancy and the postpartum period that occur to optimize fetal development, childbirth, and maternal behavior [80]. At a functional level, recent work suggests that resting-state neural activity may change from pre-pregnancy to early postpartum, with increased functional connectivity in the default mode network localized to the bilateral cuneus [81]. One reliable and valid measurement of resting-state brain activity is electroencephalogram (EEG) coherence, which provides information regarding the coordinated functioning or synchronization between brain regions [82]. A study conducted by Sandoval I.K. et al. that evaluated EEG coherence before and after giving birth demonstrated that depression scores in the postpartum period correlated with the EEG coherence values between left frontopolar and parietal areas in the alpha1 band (*r* = 0.36, *p* < 0.04). Depression symptoms were positively correlated with the EEG interhemispheric Fp1-P3 coherence measured during the postpartum period. Nevertheless, the lower Fp1-P3 coherence from pregnancy to postpartum remained when depression symptoms were controlled for in the analyses. However, the study should be considered in light of its limitations and directions for future research to assess a specific correlation between a typical EEG pattern for PPD [83].

The endeavor to diagnose PPD using predictors from both biological and EEG evaluations has increased. However, biochemicals may also serve as appropriate signs that can be employed to detect and anticipate PPD.

## 6. Treatment Options for PPD

Treatment options for PPD vary depending on the severity of symptoms, functional status, and ability to care for the newborn. Mild symptoms can be managed in primary care, while cases with psychotic symptoms, those initially unresponsive to treatment, or severe forms require emergency admission to psychiatric facilities. Thus, therapeutic strategies can be divided into two categories: non-pharmacological therapies in mild forms (psychological treatment) of PPD or pharmacological ones that can be associated with the first category (drug therapies) [84].

### 6.1. Psychotherapeutic and Psychosocial Interventions

Clinical studies have demonstrated the slight effectiveness of psychotherapy in the treatment of PPD. Valverde N. et al. conducted a systematic review regarding the psychotherapy effect for PPD in January 2023 that included seven trials with a total of 521 women who met the inclusion criteria. For the assessment of depressive symptoms, the EPDS was the most frequently used self-report symptom measurement system, applied in five of seven trials. Two studies used an author-generated questionnaire to obtain ratings on the treatment’s effectiveness. The ad hoc questionnaire assessed interpersonal connection and relationships, general well-being, outlook on life and parenting confidence, mood, and level of anxiety. Brief dynamic psychotherapy in individual format was the most frequently used type of intervention, applied in five out of seven trials, with the number of sessions ranging from four to twelve. The studies’ methodological quality was assessed considering the presence/absence of variables of the JADAD scale. The methodological analysis highlights the lack of high-quality designs concerning psychodynamic psychotherapy and PPD. Only two studies were randomized adequately out of seven; of the five with a control group, only one received treatment. The psychodynamic approach still plays a minor role in the treatment of PPD and is probably an efficient intervention. Therefore, research should be continued to assess the effectiveness of psychodynamic interventions in PPD compared to other effective treatments [85].

In another randomized trial, the goal was to compare healthcare providers trained to apply interventional psychology methods and identify patients with mild postnatal depression symptoms versus healthcare providers who did not receive this training. Thus, patients with EPDS scores ≥ 12 at six weeks postpartum were followed for 18 months. At six months postpartum, a significant number of PPD patients in the control group had higher EPDS scores compared to the two groups that received psychological interventions [86].

Qualitative research suggests that some of the barriers to treating patients with PPD include difficulties in scheduling therapy sessions and fear and stigma of dealing with a psychiatric diagnosis. Yet, studies show that they tend to choose psychological interventions instead of pharmacological treatment, especially if they are breastfeeding [54].

### 6.2. Pharmacological Antidepressive Treatments

Antidepressant therapy is recommended when symptoms do not remit with psychological therapies, when the symptomatology begins severely and requires prompt treatment, or when it is preferred by the patient [84]. Once PPD is diagnosed, prompt treatment is essential. In the absence of prompt treatment, patients are at risk of long-term disease that can lead to functional disability, worsening symptoms, resistance to treatment, and suicide [56].

The first-line antidepressants used in the treatment of PPD are selective serotonin reuptake inhibitors (SSRIs) due to their easy administration and low toxicity [54]. In a meta-analysis conducted by Zhang Q. et al. that compared the efficacy and acceptability of different pharmacotherapies for PPD, they included 11 studies with 944 participants that met the inclusion criteria. They concluded that among the most effective antidepressants was an SSRI: paroxetine (64.3%) [87]. A meta-analysis evaluating the response and remission rates of patients in the selective serotonin reuptake inhibitor (SSRI) group versus the control group demonstrated higher response and remission rates in the antidepressant group (response rate: 52.2% vs. 36.5% and remission rate: 46% vs. 25.7%) [88]. Another systematic review of randomized clinical trials demonstrates the superior efficacy of treatment with selective serotonin reuptake inhibitors compared to a placebo and (or other performed treatments) for PPD. All studies demonstrated higher response and remission rates among those treated with SSRIs and more significant mean changes on depression scales, although findings were not always statistically significant [89].

Regarding mothers with PPD who are breastfeeding, although most antidepressants pass into breast milk, most infants exposed to antidepressants while breastfeeding do not develop adverse effects. A meta-analysis of 57 studies regarding antidepressants used during breastfeeding showed reduced plasma concentrations in infants exposed to paroxetine, sertraline, or nortriptyline. However, infants exposed to fluoxetine were at a greater risk of adverse events [90]. Among the most common adverse reactions in infants of fluoxetine-treated mothers, studies have described increased crying, decreased sleep hours, gastrointestinal sensitivity, and irritability [90,91]. The maximum concentration of sertraline and fluoxetine in breast milk is reached approximately 8 h after administration. In infants of mothers on antidepressants, serum levels were below the level of quantification for each type of drug [90]. Paroxetine and sertraline are the safest selective serotonin reuptake inhibitors during breastfeeding [92].

Serotonin–norepinephrine reuptake inhibitors are used as a second-line therapy if selective serotonin reuptake inhibitors (SSRIs) are ineffective or if the patient has a history of a positive response to this type of therapeutic agent. Data regarding their use in the treatment of PPD are limited, but their efficacy has nevertheless been demonstrated in nursing mothers due to minimal passage into breast milk [84].

For patients with severe symptoms of PPD who are breastfeeding and prioritize relatively rapid improvement, the recommended antidepressant is brexanolone. Brexanolone is a neuroactive steroid synthesized by the progesterone metabolite allopregnanolone [93]. The effect of brexanolone on depressive symptoms, anxiety, and insomnia in women with PPD was analyzed in the HUMMINGBIRD clinical program, and it was demonstrated that patients receiving BRX90 (n = 102) versus a placebo (n = 107) achieved a more rapid resolution of symptoms (response rate was 81.4% vs. 67.3%) [94]. However, for patients who are breastfeeding, it is considered that they should temporarily cease nursing during treatment with brexanolone and wait until four days after the end of infusion [95]. Based on low-quality evidence, brexanolone quickly disappears from breast milk [96].

The US Food and Drug Administration (FDA) has approved zuranolone as the first oral agent indicated for PPD [97]. Zuranolone is a neuroactive steroid that functions as a GABA-A receptor-positive allosteric modulator, with a mechanism of action compatible with brexanolone, which is intravenously administered. In a phase 3 double-blind, randomized, placebo-controlled clinical trial conducted between January 2017 and December 2018, participants with diagnosed PPD were tested for the efficacity of zuranolone. Starting at day 3, remission occurred in more patients who received zuranolone than a placebo (19% vs. 5%), and remission remained greater with active medication throughout treatment [98].

In severe forms of depression, additional drug treatments are indicated. Benzodiazepines may be used temporarily for severe forms of anxiety, insomnia, or both until antidepressant medication becomes effective. Antipsychotic therapy may become necessary for forms of depression with psychotic symptoms. In severe conditions, in cases that do not respond to treatment with antidepressant agents, or in states of psychosis with suicidal ideation, hospitalization of the patient may be necessary [84].

### 6.3. Alternative Therapies for PPD

Since hormonal fluctuations are considered to be a trigger for the onset of PPD in some women, hormonal interventions have been studied in the treatment of PPD. In a study published by Dowlati et al. in 2020 regarding hormone interventions to prevent PPD, it was shown that, presently, the development of hormonal products for the prevention of PPD is at an early stage, with most trials showing preliminary, not definitive, results. Given the number of trend-level findings and the multifactorial etiology of PPD, it may be more prudent to investigate combined interventions rather than monotherapies [99].

In a similar manner, other complementary therapies used in the approach to antidepressant treatment in the postpartum period are mentioned, such as electroconvulsive therapy used in severe forms unresponsive to antidepressant medication or states with psychotic symptoms. A single more recent study conducted by Forray et al. demonstrates a 100% response rate, but data are nevertheless limited [100].

## 7. Role of Obstetrical–Gynecologist Specialists in Detecting and Preventing PPD

Women are at higher risk of developing a major depressive episode than men, and this risk is particularly accentuated by reproductive periods: adolescence, pregnancy, the postpartum period, or menopause. Female adolescents are at a two-to-threefold higher risk of major depressive disorder than males and a nearly fourfold higher risk of severe major depressive disorder. Obstetrician–gynecologists are usually the providers of medical services that women consult during these periods of vulnerability, usually with symptoms or conditions other than depression or anxiety [101].

Several depression screening methods have been validated over time, such as the Edinburgh Postnatal Depression Scale (EPDS) and the Patient Health Questionnaire (PHQ-9) [102]. The American College of Obstetricians and Gynecologists (ACOG) recommends screening by one of these scales at least once during the perinatal period [103].

Obstetricians play a unique role in identifying patients who require psychiatric evaluation in the postpartum period and could incorporate these screening methods into their practices. An ACOG’s effort should be focused on supporting by creating a more transparent environment regarding PPD screening [104]. One study conducted in the United States of America showed that the incidence of PPD symptoms was over 23% when applying screening in an obstetrical facility, higher than the overall incidence between 10% and 15% in the general population [105].

However, at the European level, there are no concrete recommendations for the integration of the assessment of a mother’s emotional status by the obstetrician and this is the main reason that women with PPD remain underdiagnosed or undertreated. Although a study conducted in Spain in 2014–2015 evaluated screening for prenatal depression during the first trimester in an obstetrics setting in a hospital in Madrid using PHQ-9, the rate of depressive symptoms was low (87.9% scored in the none-to-mild range). One study conducted in the United States of America showed that the incidence of PPD symptoms was over 23% when applying screening in an obstetrical facility, higher than the overall incidence between 10% and 15% in the general population [105]. Based on a needs assessment, researchers determined that women could be screened for PPD in the waiting room, which enabled women more privacy. The timing of the first ultrasound also allowed enough time in the prenatal period for women to participate in the 8-week group preventive intervention if they met high-risk criteria. Of the 445 women screened, 87.9% were scored in the none-to-mild risk range (PHQ-9: <9), while 9.4% were considered at high risk and recruited for the prevention study (PHQ-9: 10–14). The study had several limitations, such as the absence of incorporating screening in the third trimester. Still, the study suggests that it is feasible to integrate prenatal depression screening into obstetrics settings. There is certainly a need for more work to prevent the negative consequences associated with perinatal depression for women and their families internationally [106].

## 8. Conclusions

PPD is an increasingly common health problem among women who have recently given birth. Because of the potentially severe consequences that can affect both the patient and the family, it is crucial that healthcare providers, especially obstetricians with whom the woman comes into contact most frequently during the perinatal period, facilitate the early identification and treatment of PPD.

In addition, postnatal depression negatively affects the mother’s relationship with the infant and its development, but also the relationship with the partner and other family members. The association between molecular genetics and PPD is a highly current issue that has developed in recent years. However, there is still a small amount of research in this area, and it deserves further study due to its importance in terms of public health.

Obstetricians should include brief screening methods for PPD in the evaluation of pregnant women during their visits. Thus, if PPD is diagnosed, patients should be informed about all therapeutic techniques.

Therapeutic considerations include symptom severity, breastfeeding, and therapeutic preferences. Combined therapeutic approaches, including psychotherapy and pharmacological treatment, are recommended for treating moderate and severe forms of PPD. Complementary and alternative therapeutic methods are promising but require studies to demonstrate their effectiveness.

Until prenatal screening for PPD becomes standard practice, the healthcare system will fail to detect mothers at risk of developing depression and provide early assessment and effective treatment.

## Figures and Tables

**Table 1 diagnostics-14-00865-t001:** Genetic expression profiles associated with onset of PPF [25,33,39,40,41,42,43,44,45,46,47].

**Gene**	**Polymorphism**	**Finding**
*SERT*	*5-HTTLPR*	Increased evidence for the role played in PPD; however, studies show variable outcomes
*COMT*	*Val158Met*	Inconsistent evidence for the role played in PPD Inconsistent evidence for the role played in bipolar depression or anxiety
*OXTR*	*rs53576_CG*	Significant association with PPD after birth
*ESR1*	*TA repeat* *rs2077647*	Significant association with EPDS scores *ESR2* has not been studied in PPD
*MAOA*	*COMT-Val^158^* Met	Significant association with PPD Significant evidence for the role played in bipolar depression, negative response to antidepressant treatment, anxiety disorders, and brain activation elicited by aversive stimuli
*AKR1C2*	*SNP rs1937863*	Insignificant association with EPDS scores
*BDNF*	*BDNF Met*	Inconsistent evidence for the role played in PPD Inconsistent evidence for the role played in anxiety-like behavior in females that emerges during the period of sexual maturity
*HMNC1*	*27 single nucleotide polymorphisms*	Significant association with PPD in one study Further investigations are needed

*SERT*—serotonin transporter gene; *COMT*—catechol-O-methyltransferase gene; *OXTR*—oxytocin receptor gene; *ESR1*—estrogen receptor 1 gene; *MAOA*—monoamine oxidase A gene; *AKR1C2*—aldo-keto reductase family 1 gene; *BDNF*—brain-derived neurotrophic factor; *HMNC1*—hemicentin-1 gene; *5-HTTLPR*—serotonin-transporter-linked promoter region; *Val^158^Met*—catechol-O-methyltransferase gene functional polymorphism; *rs53576_CG*—oxytocin receptor gene polymorphism; *TA repeat; rs2077647*—estrogen receptor 1 gene polymorphisms; *SNP rs1937863*—aldo-keto reductase family 1 gene polymorphism; *BDNF Met*—brain-derived neurotrophic factor polymorphism.

## Data Availability

Not applicable.

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
