# Peer review of "Postpartum Depression: Etiology, Treatment, and Consequences for Maternal Care"

_diagnostics, 2024, doi:10.3390/diagnostics14090865_

Round 1
Reviewer 1 Report
Comments and Suggestions for Authors
Comments to the author:
This qualitative review summarizes risk factors, molecular aspects, and current detection, prevention, and interventions associated with postpartum depression (PPD). The data from this review can be used to describe the current state and contents of PPD research as well as inform potential screening and intervention methods in this vulnerable population. However, significant revisions to the organization and statistics mentioned are necessary in order to clarify the research findings. The primary concern is potentially narrowing the focus of the review, at least in terms of sections/topics under PPD explored. Otherwise the review will be unnecessarily superficial, skimming over dozens of factors related to PPD risk, detection, and treatment. Additionally, certain sentences appear randomly placed or lack relevance to the section they are in, adding little or no elaboration.
Introduction:
Line 46-49: “Recently, there has been demonstrated an increase in cases of postnatal depression, which raises the issue of an essential public health problem [7]. Postpartum depression, especially in Western countries, affects 10-15% of women in the postpartum period, even in the first year after birth [8].”
- A statistic regarding the increase of PPD would be helpful, perhaps from 2017 to a more recent year
- “From 20__ to 20__, there has been a noticeable increase in cases of PPD worldwide, from % to %, highlighting the significance of addressing this condition as a critical public health concern.”
Line 59-63: “Despite the increase in the number of postpartum depression cases in recent years, there are multiple barriers to the provision of optimal clinical care to women with this condition. Because of the social stigma of having a mental illness, most women refuse to ask for specialized help. Finally, doctors fail to catch the onset of depressive symptoms or consider them insignificant, which leads to underdiagnosis or undertreatment of the mental condition”
- Suggested edit: … the increase in PPD incidence in recent years…
- “Because of the social stigma of having a mental illness, most women refuse to ask for specialized help” could this sentence be framed differently to avoid blame language (“refuse”)?
Epidemiological factors:
Line 71-74: “The risk factors of postpartum depression are not yet fully understood. However, there is evidence that biological factors, such as hormonal factors, genetics, and immune function, among other types of causes, play an essential role in triggering postpartum depression”
Line 74: maybe rephrase to “The following sociodemographic risk factors [or social determinants of health] can be grouped according to the strength of association with PPD.”
Line 91: maybe rephrase to “Furthermore, estradiol hormone levels decrease in order to stimulate lactation. The decreased estradiol level …”
Line 108: The examples of the role of estrogen in the incidence of PPD is unclear. How does its involvement in sleep and body temperature influence PPD?
Line 111-122: **I’m not sure how I feel about this epi/SDOH part. It feels like a very rushed/random list of risk factors
Gene expression profile. Molecular aspects of PPD:
Line 133: “A recent study conducted in Sweden on 3.427 twin patients 133 and over 500.000 sisters concluded that the heritability in PPD is 54% for the first category 134 and 44% for the second category, respectively.”
Line 136: “Most of the molecular genetic studies on PPD have focused on candidate genes. These studies have selected and tested a small number of candidate genes or genetic variants for association with a phenotype.”
- Suggest switching the order of sentences in line 160 and splitting sentence in Line 164
Line 180: Positive association? Negative association?
Line 183: Suggest cutting this sentence or elaborating further
Line 186: “Genetic risk for PPD may overlap with genetic risk for other types of psychiatric 186 conditions”
- List the other psychiatric conditions
- The psychiatric conditions in question were not really talked about in the previous paragraphs. Elaborating in the previous paragraphs on which genes overlap with which other psychiatric conditions would be helpful
Line 187: Why are these studies challenging to perform?
Line 188: “it”? Is this referring to PPD or the candidate gene association studies?
Clinical presentation of PPD onset:
Line 229: The relevance of psychiatric history is not mentioned in this section?
**Clinical presentation seems a bit rushed. Social stigma and underdiagnosis are not mentioned despite emphasis in the abstract. Maybe this could be combined with the next section?
Identification and screening for women at risk for PPD:
Cite 52-53?
The concluding paragraph of this section is a great explanation of current controversies and developing interventions.
Treatment options for PPD:
Line 307: the organization of this paragraph and introduction into the subsections was concise and well-done
Line 319: “This is a viable treatment method for mild forms of postpartum depression, espe- 319 cially for breastfeeding mothers who refuse pharmacological treatment.”
- Why breastfeeding mothers in particular?
- Maybe framing other than “refuse”? Or add a reason for “refusal”?
Line 330: “He concluded that all studies reported the efficacy of psychodynamic interven- 330 tions for PPD in both home and clinical settings and in group and individual formats”
- This does not describe how “clinical studies have demonstrated the effectiveness of psychotherapy”
- A statistic of efficacy/effectiveness would be helpful
Line 347: elaborate on the difference in patient tolerance for psychological interventions vs pharmacological treatment
Line 351: what is the 'usual medical assessment'?
Line 368: In this context, does “potent” mean “effective”?
Line 382: greater risk for adverse effects?
Role of OBGYN in detecting and preventing PPD:
Line 455: What was the incidence statistic expected?
Line 458: “Obstetrical complications associated with birth defects could significantly impact
maternal emotional status. In a study conducted in Romania, the risk of congenital heart
disease was significantly increased in the ART group compared to the spontaneous conception group”
- The statistic does not connect to the previous sentence – how does risk of congenital heart disease in ART vs spontaneous conception apply to maternal emotional state, specifically PPD?
Line 464: what is “most of the time”?
Line 470: Specifying preventative measures would be helpful here, particularly regarding the focus of the section: obstetric screening
Conclusions:
Line 480 can be removed: repetitive of Line 475
The conclusion is concise but could use reorganization in terms of summarizing the paper and highlighting the takeaways of each section.
Comments on the Quality of English Language
n/a
Reviewer 2 Report
Comments and Suggestions for Authors
The general review is informative and provides an adequate update on PPD in terms of epidemiology, biology, and treatment. However, the authors left out two medications for PPD recently approved by the US FDA for PPD, namely brexanolone and zuranolone, one being allopregnanolone and the other its derivative, that are positive allosteric modulators of the GABAA receptor. Given these drugs, the authors should include some details on genes that are involved in neuroactive steroid biology (progesterone and allopregnanolone) and include these two medications as treatments along with the other antidepressants mentioned by the authors.
Reviewer 3 Report
Comments and Suggestions for Authors
This manuscript addresses an important issue but requires significant modifications.
I suggest shortening some sections and adding others so that it adds something new to the current state of knowledge
Major concerns:
The section about the treatment (medication, psychotherapy and alternative options is too long
Diagnostic should be the basis and the key point of work
there is also no chapter on other possibilities when it comes to diagnostics: biological tests, blood biomarkers, or EEG tests
this section should be added and described in detail
Authors mentioned the gene SERT as an engaged in the PPD (lines 143-146) "in line 146: however, studies show variable outcomes" - Authors should cite the original papers (instead of 33:
r concerns:Payne, J.L. Genetic basis for postpartum depression. Biomarkers for postpartum psychiatric disorders. 2019;15-34)
The information about genetic factors engaged in PDD (Section 3) should be implemented in the figure such as:
https://www.mdpi.com/1424-8247/14/3/204 (Figure 1). I suggest replacing table with well-known diagnostic criteria of PDD by more interesting and a novel table with genetic factors
See:
https://www.ncbi.nlm.nih.gov/pmc/articles/PMC6059965/
https://www.sciencedirect.com/science/article/pii/S0165032722008990
https://europepmc.org/article/med/30808206
https://www.sciencedirect.com/science/article/pii/S0165032717318165
Some minor issues:
Authors should revise the abbreviations PDD/PPD in the abstract.
DSM-V: please change to the proper name of the scale DSM-5
In the first sentence of the main text, authors using abbreviations, and in the text of the manuscript, this should be used consequently instead of other names such as Postpartum depression
Line 52: second-time authors define the abbreviation - please remove it
In the last sentence of the background, the aim of the study should be highlight
In the first sentence of the second section, authors should classify the risk factors - divided into some of the subgroups
Lines 117-118: The authors mentioned food products that decrease the risk of PPD. Please add information on which types of foods potentially increase risk.
Line 120-121 "Periods of severe sleep deprivation or chronic sleep deprivation affect carbohydrate metabolism, inflammatory processes, mental health, and quality of life"
please explain how sleep patterns affected the aforementioned
Line 214: the Authors indicated The postpartum blues and psychosis but did not mention how DSM-5 refers to these states. Please describe where these state are in the DSM-5 criteria.
Round 2
Reviewer 3 Report
Comments and Suggestions for Authors
The authors carefully revised the manuscript. I have no doubt.
Some minor mistakes:
Table 1 needs legend and references
DSM-4 should be changed to DSM-IV
Author Response
1)Table 1 needs legend and references
Response: The references has been added at the end of Table 1 title: [25] [33] [39-47]
Legend
*SERT- serotonin transporter gene
*COMT- catechol-O-methyltransferase gene
*OXTR- oxytocin receptor gene
*ESR1- estrogen receptor 1 gene
*MAOA- monoamine oxidase A gene
*AKR1C2- aldo-keto reductase family 1 gene
*BDNF- brain-derived neurotrophic factor
*HMNC1- hemicentin-1 gene
*5-HTTLPR- serotonin-transporter-linked promoter region
*Val158Met- catechol-O-methyltransferase gene functional polymorphism
*rs53576_CG- oxytocin receptor gene polymorphism
*TA repeat; rs2077647- estrogen receptor 1 gene polymorphisms
*SNP rs1937863- aldo-keto reductase family 1 gene polymorphism
*BDNF Met- brain-derived neurotrophic factor polymorphism
2) DSM-4 should be changed in DSM-IV
Response: DSM-4 has been changed to DSM-IV. The information can be found on page 6, fourth paragraph.